# Cloaking nanoparticles with protein corona shield for targeted drug delivery

Jun Yong Oh[1], Han Sol Kim[2], L. Palanikumar[1], Eun Min Go[3], Batakrishna Jana[1], Soo Ah Park[4], Ho Young Kim[3], Kibeom Kim[1], Jeong Kon Seo[2], Sang Kyu Kwak [3], Chaekyu Kim [1], Sebyung Kang[2] & Ja-Hyoung Ryu [1]

Targeted drug delivery using nanoparticles can minimize the side effects of conventional pharmaceutical agents and enhance their efficacy. However, translating nanoparticle-based agents into clinical applications still remains a challenge due to the difficulty in regulating interactions on the interfaces between nanoparticles and biological systems. Here, we present a targeting strategy for nanoparticles incorporated with a supramolecularly pre-coated recombinant fusion protein in which HER2-binding affibody combines with glutathione-S-transferase. Once thermodynamically stabilized in preferred orientations on the nanoparticles, the adsorbed fusion proteins as a corona minimize interactions with serum proteins to prevent the clearance of nanoparticles by macrophages, while ensuring systematic targeting functions in vitro and in vivo. This study provides insight into the use of the supramolecularly built protein corona shield as a targeting agent through regulating the interfaces between nanoparticles and biological systems.

[1] Department of Chemistry, Ulsan National Institute of Science and Technology (UNIST), Ulsan 44919, Republic of Korea. [2] Department of Biological Sciences, Ulsan National Institute of Science and Technology (UNIST), Ulsan 44919, Republic of Korea. [3] Department of Energy Engineering, School of Energy and Chemical Engineering, Ulsan National Institute of Science and Technology (UNIST), Ulsan 44919, Republic of Korea. [4] In Vivo Research Center, UNIST, Central Research Facilities, Ulsan National Institute of Science and Technology (UNIST), Ulsan 44919, Republic of Korea. These authors contributed equally: Jun Yong Oh, Han Sol Kim, L. Palanikumar. Correspondence and requests for materials should be addressed to C.K. (email: chaekyu@unist.ac.kr) or to S.K. (email: sabsab7@unist.ac.kr) or to J.-H.R. (email: jhryu@unist.ac.kr)

The application of nanoparticles is promising for the development of imaging and therapeutic agents through improved biodistribution and controlled drug release[1–3]. The rationale behind using nanoparticles is that those with diameters of <200 nm extravasate from the leaky tumor blood vessels are retained in the tumor due to the "enhanced permeability and retention" (EPR) effect[4]. Although few nanoparticle formulations (e.g., Abraxane and Doxil) are available in the market, the ubiquitous targeting approach suffers from several limitations including rapid clearance by the mononuclear phagocyte system (MPS) and low uptake into target tumors[5,6]. To improve the targeting ability, along with the EPR effect, an active targeting approach has been attempted by coating the particle surface with antibodies, proteins, or peptides that bind to receptors that are typically overexpressed on cancer cells[7]. However, recent reports have revealed only a modest increase in tumor targeting when this approach is applied; moreover, the addition of targeting ligands increased the clearance of nanoparticles by MPS, indicating that no definitive conclusion has been reached regarding the therapeutic efficacy of this technique[8,9].

In principle, when exposed to physiological environments, the nanoparticle surface is covered by various biomolecules to lower the surface energy by a combination of entropy-driven water molecule displacement, particle surface charge compensation, and screening of hydrophobic parts[10–12]. Biomolecule adsorption results in the formation of a layer, called a protein corona, and significantly changes the original molecular identity of the nanoparticle[13]. The formation of a protein corona on the nanoparticle surface can be regulated by modifying the nanoparticle surface with zwitterions, polyethylene glycol (PEG), carbohydrate moieties, and dysopsonic proteins, which can enhance the colloidal stability and prolong the circulation time in blood by enabling escape from MPS clearance[14–17]. However, these strategies are still limited at conferring targeting specificity since an additional targeting ligand increases the propensity for protein corona formation to mask the targeting ability and inhibit the biological effects of nanoparticles[18–24]. This can explain why many nanoparticles with active targeting systems have failed in clinical trials[25]. Therefore, to design nanoparticle-based therapeutic agents, there is a considerable need to regulate protein corona formation on nanoparticles[26–28] and to obtain a deeper understanding of the molecular mechanism involved in regulating nanoparticle–biological interactions. Here, we present a targeting system in which nanoparticles are supramolecularly pre-coated with a protein corona shield (PCS) that reduce serum protein absorption while retaining targeting specificity (Fig. 1a).

## Results and Discussion

### The development of PCS on nanoparticle.
To develop PCS on nanoparticles, we use a recombinant fusion protein, GST-HER2-Afb, in which HER2-binding affibody (Afb) is genetically combined with a glutathione-S-transferase (GST)[29], a well-known fusion tag protein, with an extra linker (GGGLVPRGSGGGCGGGGTGGGSGGG). The preparation of GST-HER2-Afb (molecular weight: 36.3 kDa, >99.0% purity) is confirmed by electrospray ionization time-of-flight mass spectrometry (Fig. 1b) and sodium dodecyl sulfate–polyacrylamide gel electrophoresis (SDS-PAGE) (Supplementary Fig. 1A). The surface charge of the GST-HER2-Afb at physiological pH is approximately −5.25 mV, which is similar to that of GST (−6.29 mV) (Fig. 1c). The intact binding ability of GST-HER2-Afb is confirmed by monitoring its interactions with the targeting receptor HER2/ErbB2 in real time using a quartz crystal microbalance (Supplementary Fig. 1B) and the cellular uptake of the fluorescein-labeled GST-HER2-Afb to the

HER2-receptor-overexpressing cell line (SK-BR3) (Fig. 1e). These results indicate that the recombinant fusion protein with the linker, GST-HER2-Afb, exhibit the ability to bind to the complementary receptor. Furthermore, the toxicity test confirm its biocompatibility, indicating that GST-HER2 itself is not toxic up to 10 μM (Supplementary Fig. 1C).

Next, PCS nanoparticles (PCSNs) were constructed by supramolecularly attaching GST-HER2-Afb to mesoporous silica nanoparticle (MSN), for which cargo molecules can be loaded in the interior (Fig. 1a)[30]. First, 3-(trimethoxysilyl) propyl acrylate was modified on MSN (mean diameter: 140 ± 10 nm in a dynamic light scattering (DLS), surface area: 1190 m$^2$/g, pore volume: 1.10 cm$^3$/g, mean pore size: 2.68 nm; Supplementary Fig. 2A–C; Supplementary Table 1) and glutathione (GSH) was further attached using thiol-ene click chemistry, which was confirmed by FT-IR analysis (Supplementary Fig. 2D). The hydrodynamic radius of the GSH-modified MSN (GSH-MSN) was ~140 ± 20 nm, as confirmed by DLS experiment, and the diameter of GSH-MSN observed from transmission electron microscopic (TEM) images was ~90 ± 10 nm (Fig. 1f). GSH modification decreased the surface area (540 m$^2$/g), pore volume (0.5 cm$^3$/g), and mean pore size (2.04 nm) (Supplementary Fig. 2B, C; Supplementary Table 1). Nevertheless, the GSH-MSN showed a decent cargo loading capacity of 52%, 65%, 11%, and 5% for doxorubicin (DOX), camptothecin (CPT), DiIC$_{18}$ (DiI), and DiD, respectively (Supplementary Fig. 3; Supplementary Table 2).

The GSH-MSN was then coated with GST-HER2-Afb to give the PCSN through the supramolecular interaction at the GSH-binding site. The attachment of GST-HER2 on GSH-MSN was confirmed by measuring the surface charge, which changed from −40 mV (GSH-MSN) to −5.3 mV (PCSN) (Fig. 1c). The DLS experiment indicated an increase in the hydrodynamic radius of PCSN (~270 ± 20 nm) compared with that of GSH-MSN (~140 ± 20 nm) (Supplementary Fig. 4A), and the TEM image analysis further confirmed the protein-coating layer (Fig. 1f). The maximum number of GST-HER2-Afb attached on GSH-MSN was 270 μg/mg, which was confirmed by BCA assay (Supplementary Fig. 4B). No significant aggregation of PCSNs was observed in the DLS analysis up to 2 weeks at 4 °C (Fig. 1d).

### The interaction between PCSN and serum proteins.
Next, we investigated interactions occurring at the interfaces between serum proteins and PCSNs. Control particles, GSH-MSN, and PEG-modified MSN (PEG-MSN)[31] were used to study the effects of GST-HER2-Afb pre-coating on particles. We first incubated PCSN, PEG-MSN, and GSH-MSN with 55% fetal bovine serum (FBS) for 1, 2, and 4 h and isolated the serum proteins that adsorbed to them by centrifugation to completely remove unbound proteins. The molecular composition of the adsorbed serum proteins was measured by denaturing SDS-PAGE and the protein density was plotted (Fig. 2a, b). The protein profiles observed for PEG-MSN and GSH-MSN were fairly similar, but the intensities of the bands for PCSN were significantly reduced (~15-fold lower intensity than for GSH-MSN for the case of 1 h of incubation), indicating that the GST-HER2-Afb pre-coating on PCSN could reduce the interactions with serum proteins. To further confirm the particle–protein interactions, the physical characterizations were also investigated by using DSL, zeta, and PDI (Supplementary Fig. 5). The treatment of 55% serum on GSH-MSN and PEG-MSN increased its hydrodynamic radius and changed its surface charge while PCSN showed no significant changes, confirming the decreased PCSN–serum protein interactions.

We then investigated the composition of serum proteins adsorbed on each particle using shotgun proteomics [liquid

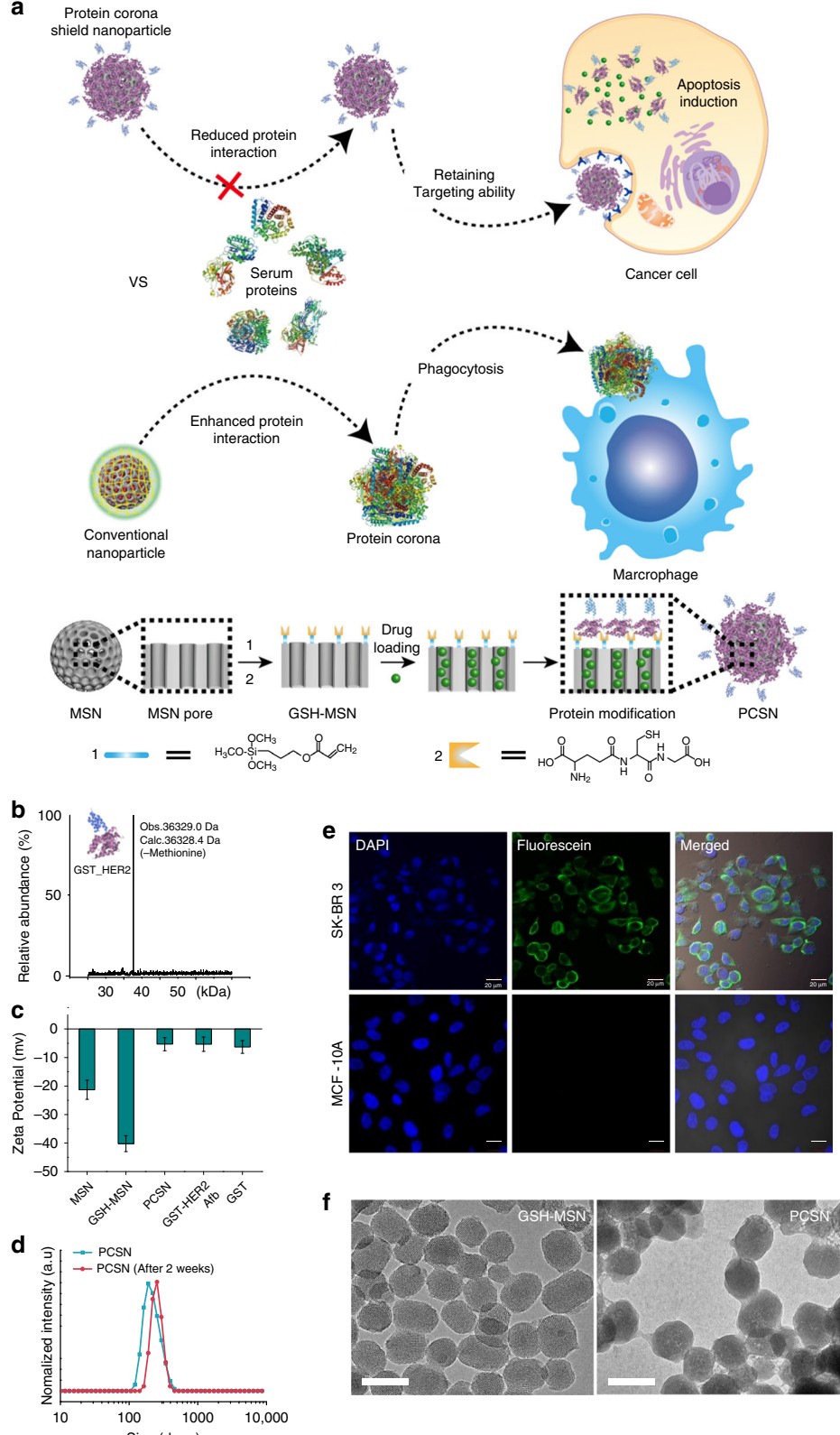

chromatography tandem mass spectrometry (LC-MS/MS)]. A total of 183 proteins were identified and the 78 most abundant proteins (at levels >0.01% w/w) were selected and divided into three groups based on the correlation in their relative abundances on each particle (Group 1: GSH-MSN > PEG-MSN > PCSN,

Group 2: GSH-MSN ≈ PEG-MSN > PCSN, Group 3: GSH ≈ PEG-MSN ≈ PCSN, as shown in Fig. 2c), which exhibited the lowest tendency to be adsorbed on PCSN. The proteins were further classified according to the weight, isoelectric point, and serum protein classification (expressed as a percentage of each

**Fig. 1** Protein corona shield nanoparticle (PCSN). **a** We introduce the protein corona shield (PCS) concept for an efficient target drug delivery system. Generally, nanoparticle drug carriers with a target ligand lose their targeting ability on being coated by blood proteins in a biological environment. However, the PCS system can inhibit blood protein adsorption to maintain the targeting ability and avoid unwanted clearance by the mononuclear phagocyte system. **b** Mass spectrometry analysis of the GST-HER2-Afb showed a mass of 36.3 kDa. **c** Zeta-potential analysis of mesoporous silica nanoparticle (MSN) (−23 mV), GSH-MSN (−39 mV), GST-HER2-Afb (−5.25 mV), and PCSN (−5.3 mV). **d** Size distribution plots of PCSN. **e** Images of cellular uptake of fluorescein 5 maleimide-modified GST-HER2-Afb by the target cell (SK-BR3) and the negative control (MCF-10A). **f** Transmission electron microscopic images of GSH-MSN and PCSN (scale bar represents 100 nm). All bar graphs were reported as means ± standard deviations (SDs) for three experimental replicates ($n = 3$)

protein in Fig. 2d, middle). The proteins adsorbed on PCSN were composed of lower-molecular-weight proteins (20–60 kDa) (Fig. 2d, left) and included an abundance of proteins with pI of approximately 7–8 compared with the proteins adsorbed on GSH-MSN and PEG-MSN (Fig. 2d, right). Moreover, the amount of immune response involving proteins, complement, and coagulation proteins adsorbed on PCSN was significantly lower than those on GSH-MSN and PEG-MSN, suggesting that PCSN can increase their circulation time in the blood by decreasing the uptake from the immune system[20]. It is also noted that the relative quantity of apolipoproteins among serum proteins absorbed on PCSN increased with respect to GSH-MSN and to PEG-MSN (Fig. 2d, middle). Considering that apolipoproteins on nanoparticle play an important role in association with cellular uptake to target cells[32–34], targeting capability of PCSN partly could be linked with some serum proteins recruited at the surface of PCSN.

To understand the interactions between PCS and external biological components, we investigated the molecular mechanisms of the interactions associated with the conformational change of GST-HER2-Afb. When non-covalently adsorbed on nanoparticles with precisely designed supramolecular interaction, specific domains of proteins can interact with the surface of nanoparticles in thermodynamically favorable manners, leading to the colloidal stabilization and dysopsonization of particles in a physiological environment[35,36]. On the other hand, covalent attachment of their counterparts on particles significantly changes their conformations and particle surface properties (e.g., hydrophobicity and charge density), inducing potential detrimental interactions with external biomolecules, rather than stabilizing them[13,19,37]. Therefore, assuming that the orientation and conformational change of proteins adsorbed on particles are subject to interaction with serum proteins, the effects of chemical modification of GST-HER2-Afb on biological consequences were investigated. First, PCSNs having randomly oriented GST-HER2-Afb [PCSN(R)s] were prepared by chemically conjugating GST-HER2-Afb to succinimidyl-modified MSN via amide formation with amine groups on the protein surface (Fig. 2e, right). When assessed by SDS-PAGE (Fig. 2e, left), the protein absorption for PCSN(R) was enhanced five-fold compared with that for PCSN, indicating that the randomly orientated conjugation of GST-HER2-Afb may change surface properties and increase protein–nanoparticle interactions. Next, the succinic anhydride-modified PCSN [PCSN(−)] were prepared to generate additional carboxylate groups on GST-HER2-Afb, which has a negatively charged function but still binds on GSH-MSN in the same orientation as GST-HER2-Afb (Fig. 2e, middle). The SDS-PAGE analysis confirmed that the protein absorption observed for PCSN(−) was similar to that of PCSN, implying that once thermodynamically favorably orientated on nanoparticles, the supramolecularly adsorbed proteins can stabilize the colloid and significantly reduce the interactions with external serum proteins. We further employed coarse-grained

molecular dynamics method to observe the interactions between coated silica nanoparticles (i.e. PEG-MSN, PCSN(R), PCSN (−), and PCSN) and serum protein (i.e. albumin) in the vicinity of their interface at the molecular level (simulation method details in the Supplementary Figs. 6–12), which suggests that the orientation of GST-HER2-Afb attached to the surface of silica nanoparticles as well as the electrostatic interactions can be an important factor.

**Stealth and targeting effect of PCSN.** We studied the cellular uptake of PCSN by macrophages, one of the most important components of the immune defense system, which act by clearing foreign molecules from the blood. Being able to evade internalization by phagocytic cells would provide a drug carrier that can accomplish long-term blood circulation and enhanced arrival at the target tumor[38,39]. PCSNs or PEG-MSNs loaded with a fluorescent dye, DiI, were pretreated with 55% FBS for 1 h at 37 °C and then incubated with a murine macrophage-like cell line, RAW264.7, for 6 h in culture medium supplemented with 10% serum (Fig. 3a). Confocal microscopy imaging revealed that the internalization of the DiI-loaded PEG-MSN was significant, but was rarely observed for the DiI-loaded PCSN (Fig. 3b), which was further confirmed by flow cytometry (Fig. 3c). Cell viability analysis also confirmed reduced cellular uptake of PCSN when the camptothecin-loaded PCSN was applied (Fig. 3d). These results indicate that PCS on PCSN significantly reduced internalization into macrophages. A recent study reported that the biomolecular corona formation with a specific composition provides a stealth effect on the macrophage recognition[40]. Similarly, our results suggest that the scarce corona formation on PCSN caused by the supramolecularly pre-coated proteins could confer stealth properties to evade immune cells and subsequent elimination by the MPS.

Next, to investigate the cell-specific targeting ability of PCSN, the HER2-receptor-overexpressing cancer cell line SK-BR3 and the HER2-receptor-negative cell line HEK293T were treated with Dil- or camptothecin-loaded PCSN after pre-incubation in 55% FBS for 4 h (Fig. 3e). The significant uptake of Dil-loaded PCSN by SK-BR3 cells rather than HEK293T cells, mediated by receptor-mediated endocytosis (especially macropinocytosis), was confirmed by confocal fluorescence microscopy (Fig. 3f and Supplementary Fig. 13A) and FACS (Supplementary Fig. 13B). The targeted internalization was further confirmed by measuring cellular viability for the camptothecin-loaded PCSN, which exhibited dose-dependent toxicity on SK-BR3, while exhibiting lower toxicity on HEK293T (Fig. 3g). A control, HER2-binding affibody (Afb) modified MSNs (Afb-MSN) was also evaluated and it confirmed that GST plays an important role in reducing interactions among the serum proteins as well as supramolecularly conjugating on particle (Supplementary Fig. 14). To examine the versatile targeting platform for PCS, EGFR-binding Afb combined with glutathione-S-transferase (GST-EGFR-Afb) was also applied on GSH-MSN and its targeting ability to MDA-MB468 cells (EGFR-positive cancer cells) was confirmed

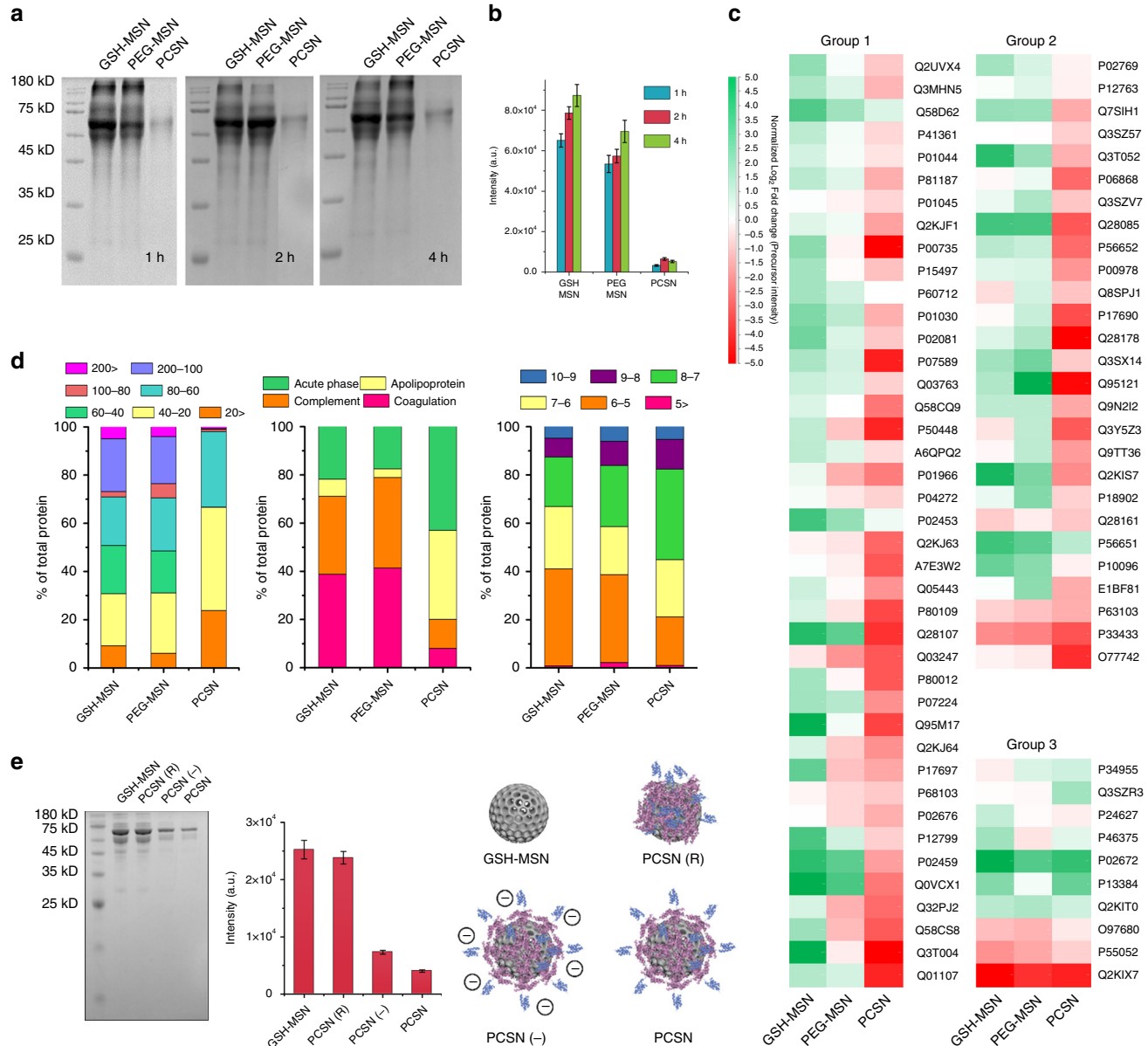

**Fig. 2** Proteomic study of surface protein corona. **a** GSH-MSN, PEG-MSN, and PCSN were treated with 55% serum for 1, 2, and 4 h, and the amount of serum protein attached to the surface was determined by SDS-PAGE. **b** Band intensity difference. **c** Classification of protein corona components characterized by quantitative LC-MS/MS. A total of 183 proteins were identified and the 78 most abundant proteins were used to make the heat map. **d** Proteins attached to each particle were classified by weight (kDa), category, and pI. **e** GSH-MSN, PCSN(R), PCSN(−), and PCSN were treated with 55% serum for 1 h and the amount of serum protein attached to the surface was determined by SDS-PAGE. All bar graphs were reported as means ± standard deviations (SDs) for three experimental replicates (n = 3)

(Supplementary Fig. 15). Taken together, these results show that PCSN exhibited cell-specific targeting ability as well as stealth properties.

**Effect of PCSN on the targeting and antitumor efficacy.** To translate the outcomes of PCSN in an in vivo tumor model, nanoparticles loaded with the far-red fluorescent dye DiD were systemically injected via the tail vein into nude mice (n = 6 mice per group) bearing SK-BR3 cell xenografts, after which the tumor accumulation of PCSN was monitored. Considering the lowest accumulation efficacy of GSH-MSN observed from initial experiments (Supplementary Fig. 16A), our studies focused on the comparison between PCSN and PEG-MSN. The in vivo live

imaging results showed the presence of 1.8-fold enhanced fluorescent signals in tumor sites from the PCSN-treated group compared with the PEG-MSN-treated group at 0.5 h after injection (Supplementary Fig. 16B), which was maintained until 24 h, while the fluorescent signals for PEG-MSN gradually decreased with time at 8 h (Supplementary Fig. 16C).

The biodistribution of nanoparticles was then assessed by measuring the fluorescence of DiD in harvested organs and tumors upon necropsy 48 h after injection. The fluorescent signals of tumors from the PCSN group were enhanced 2.5-fold in comparison with those of the PEG-MSN group (Fig. 4a). Moreover, the fluorescent intensity in the tumor for the PCSN group was seven-fold higher than that in reticuloendothelial organs (e.g., liver and spleen), whereas no significant difference in

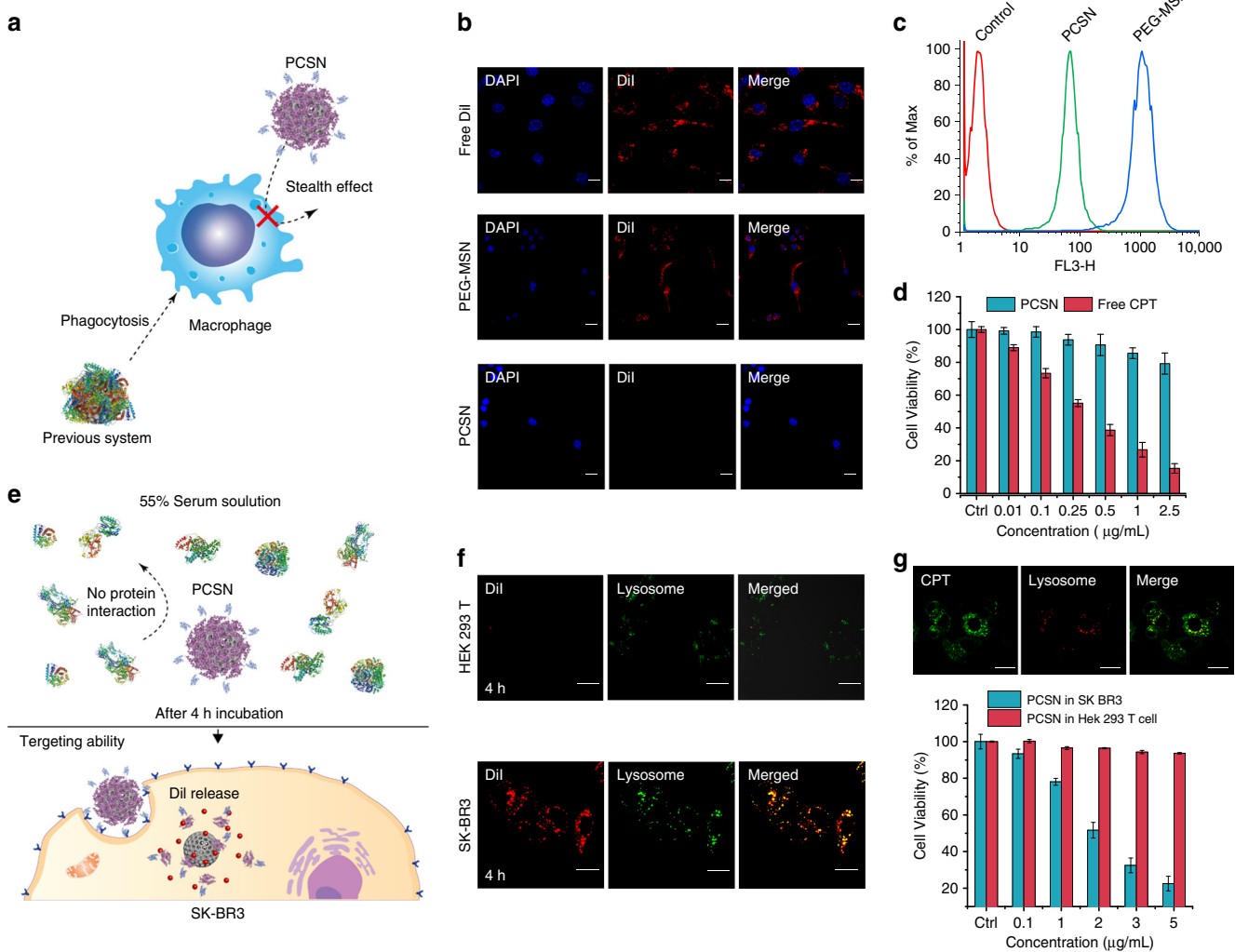

**Fig. 3** In vitro experiment and stealth effect of PCSN. **a** Schematic showing the avoidance of phagocytosis by a macrophage. **b** Confocal microscopy images of DiI-loaded PCSN and PEG-MSN, and free DiI incubated for 6 h in RAW264.7 cells (scale bar is 20 μm). **c** FACS analysis of PCSN and PEG-MSN incubated in RAW264.7 cells for 6 h. **d** Cell cytotoxicity assay of PCSN and free camptothecin (CPT) on RAW264.7 cells (48 h of incubation). **e** Schematic of the targeting ability of PCSN treated with 55% serum. Cellular uptake confocal microscopy images of **f** DiI-loaded PCSN to HEK293T cells (negative control) and to SK-BR3 (target cells). **g** Cellular uptake confocal microscopic images of camptothecin-loaded PCSN to SK-BR3 and cell cytotoxicity assay (scale bar is 10 μm). All bar graphs were reported as means ± standard deviations (SDs) for three experimental replicates ($n = 3$)

fluorescent intensities between tumors and the reticuloendothelial organs was observed for the PEG-MSN groups (Fig. 4b), indicating that PCS enabled the nanoparticles to evade the immune system and undergo enhanced accumulation in the target tumor. In vivo antitumor efficacy was further evaluated by intravenously administering camptothecin-loaded PCSN (PCSN (CPT)), PEG-MSN (PEG-MSN(CPT)), PCSN, camptothecin (Free-CPT), and phosphate-buffered saline (PBS) to the SK-BR3 tumor-bearing mice. Compared with the PEG-MSN(CPT)-treated group, the PCSN(CPT)-treated group exhibited higher therapeutic efficacy, resulting in ~90.0% inhibition of tumor growth in terms of volume and 2.5-fold enhancement of inhibitory effects (Fig. 4c). Subsequently, by examining tumor weight reduction and histopathology after necropsy on day 22, the PCSN(CPT)-treated group was shown to have enhanced therapeutic efficacy for tumor growth inhibition (Supplementary Fig. 16D, 17). Additionally, hematoxylin and eosin (H&E)-stained lung, liver, spleen, and kidney samples showed no apparent abnormalities or lesions at day 21 after camptothecin-loaded PCSN treatment (Supplementary Fig. 17). These results indicate

that PCSN increases the tumor targeting ability, enhancing the efficacy of cancer chemotherapy. Collectively, we found that the supramolecular binding of GST-HER2-Afb on particles enabled the maintenance of conformational stability and further minimized interactions with serum proteins. In vitro and in vivo experiments confirmed that PCSN improved the targeting ability and therapeutic efficacy. These findings indicate that exploiting protein coronas can provide a tool for a targeting platform.

## Methods

**Materials.** Cetyltrimethylammonium bromide (CTAB), Pluronic® F-127 ($EO_{106}PO_{70}EO_{106}$), tetraethyl orthosilicate (TEOS; 98%, reagent grade), fluorescein isothiocyanate, toluene (99.9% anhydrous), pyridine, dimethylformamide (HPLC grade), disuccinimidyl suberate, and dimethyl sulfoxide were purchased from Sigma-Aldrich (Yongin S. Korea). 3-(Trimethoxysilyl) propyl acrylate, (3-amino-propyl) trimethoxysilane, succinic acid anhydride, and L-glutathione reduced (>98%) were purchased from Tokyo Chemical Industries (TCI) (Tokyo, Japan). 1,1′-Dioctadecyl-3,3,3′,3′-tetramethylindocarbocyanine perchlorate (DiI), DiD were purchased from Thermo Fisher. CPT was obtained from Ontario Chemicals Inc. (Ontario, Canada). Doxorubicin (Dox) was obtained from Acorn PharmaTech (Redwood City, CA, USA). Anhydrous ethanol, sodium hydroxide (NaOH; 99%), and ammonium hydroxide (29 wt%) were purchased from Samchun Chemical. All

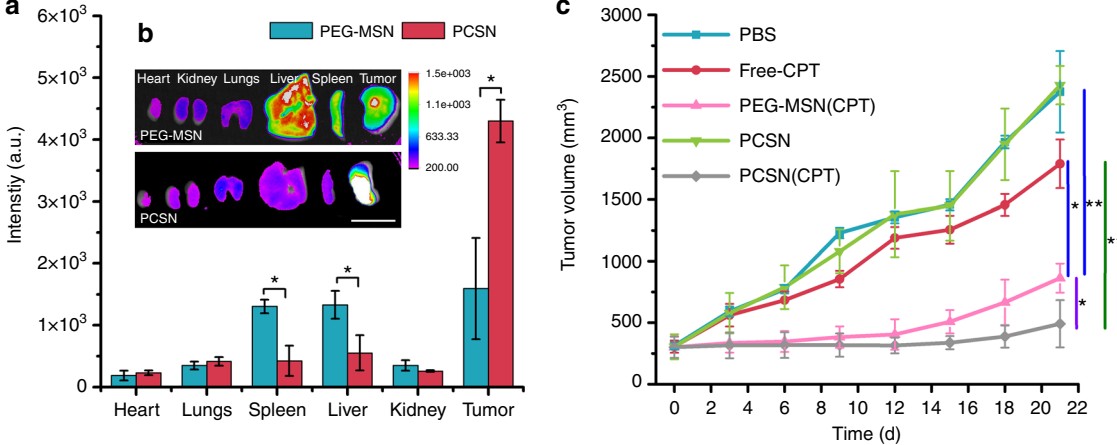

**Fig. 4** Ex vivo and in vivo efficiency of PCSN. **a**, **b** Fluorescence images of organs and tumors 48 h after intravenous injection and biodistribution of injected formulations in animals with SK-BR3 tumor xenograft from fluorescence intensity analysis. In vivo antitumor effects in different treatment groups loaded with camptothecin (CPT) (1.5 mg/kg of mice) (scale bar is 2 cm). **c** Growth curve of tumor volume after intravenous injection with various groups of carriers until day 21 ($n = 6$ mice per group, mean ± 1 day [$n = 6$ mice per group, mean ± SD, statistical significance was calculated by one-way analysis of variance, *$P < 0.05$, **$P < 0.01$]

chemicals were used as received without further purification. Deionized (DI) water was produced by the Millipore Milli-Q System (18.2 MΩ cm). Isopropyl β-D-1-thiogalactopyranoside was purchased from Bio Basic. Lysozyme was purchased from Sigma-Aldrich. PBS (10×) without calcium or magnesium was purchased from Lonza. One milliliter HisTrap FF column was purchased from GE Health-Care. DuoflowTM chromatography system was purchased from Bio-rad. BCA assay was purchased from Thermo Fisher. MassPREP microdesalting column and Xevo G2 TOF MS were purchased from Waters. All the other chemicals were purchased from Bioshop. SK-BR-3 cells were purchased from the Korean cell line bank (KCLB catalog No. 30030). MDA-MB-468 cells were purchased from the Amerian Type Culuture Collection (ATCC Catalog No. HTB-132). All cell culture reagents and medium were from Life Technologies (S. Korea) and FBS was purchased from Gold Standard (USA). The graphics found in the figures were created by a co-author.

**Cell culture**. Human breast cancer cells derived from the metastatic sites MDA-MB468 and SK-BR3 were obtained. Normal macrophage cell lines RAW264.7 and HeK293T were obtained as a gift from Prof. Hyun Woo Rhee at UNIST. SK-BR3 and RAW264.7 cells were cultured in DMEM medium (Invitrogen, S. Korea), and MDA-MB468 in Leibovitz-L-15 media (Invitrogen, S. Korea) was supplemented with 10% FBS, 100 µg/mL streptomycin, and 100 U/mL penicillin at 37 °C in a humidified incubator containing 5% CO$_2$ and 95% air. The medium was replenished every other day, and the cells were subcultured after reaching >85% confluence.

**Cell viability analysis**. The in vitro cell viabilities of various formulations against SK-BR3 and MDA-MB468 cells were determined by performing the Alamar blue dye assay (Invitrogen, Korea). Briefly, SK-BR3 and MDA-MB468 cells were cultured in 96-well (Thermo Scientific Inc. Korea) micro-titer plates at a density of $5 \times 10^3$ cells/well and then allowed to settle for 24 h under incubation at 37 °C, 95% air, and 5% CO$_2$. Then, the grown cells were treated with different concentrations of pristine MSN, GSH-MSN, and EGFR- and Her2-Afb-modified MSN (10 µg/mL to 1 mg/mL) in both cell lines and analyzed after 24-h incubation at an excitation wavelength of 565 nm and an emission wavelength of 590 nm using a fluorescence plate reader (Tecan Infinite Series, Germany). Similarly, CPT-loaded EGFR and Her2 PCSN along with free-CPT were analyzed using similar methods (0.01, 0.1, 0.25, 0.5, 1, 2.5, and 5 µg/mL of CPT).

**Cellular uptake analysis**. The cellular uptake of CPT-loaded nanoparticles was investigated with confocal microscopic and flow cytometric analyses. HeK293T, MDA-MB468, and SK-BR3 cells were seeded into two-well chambers with a cover glass (Lab Tek II; Thermo Scientific) at a seeding density of $2 \times 10^5$ cells/well. After a 24-h incubation, the cells were treated with CPT-loaded PCSN at a final concentration of CPT of 10 µg/mL at different time points and analyzed with confocal microscopy. Similarly, the cells were stained with lysotracker green (Lysotracker Green FM DND-26; Invitrogen) to check the colocalization with the CPT-loaded MSN. To evaluate the drug release from MSN, the FITC-conjugated MSNs were loaded with CPT and the releases at varying time points (2, 1, and 16 h) were checked. To evaluate cellular uptake, DiI-loaded MSNs were used for flow

cytometric analysis. HeK293T, SK-BR3, and MDA-MB468 cells were seeded into six-well plates at a density of $1 \times 10^6$ cells per well and incubated in a complete medium for 24 h at 37 °C, 95% air, and 5% CO$_2$. The concentration of DiI nanoparticles was equivalent to a DiI dosage of 0.20 µg/mL. After the stipulated period of 4 and 6 h of incubation, the different treatment cells were trypsinized, harvested, rinsed with PBS, resuspended, and subjected to flow cytometry assay using BD-FACS Caliber. All experiments detected ≥10,000 cells, and the data were analyzed using the FlowJo software.

**Endocytic pathway analysis**. To check the endocytosis-mediated uptake, SK-BR3 cells were seeded into four-well chambers with cover glass and pretreated with different inhibitors, including sucrose (clathrin-mediated uptake, 400 nM), methyl-beta cyclodextrin (caveolae-mediated uptake), and amilorin (macropinocytosis), in a serum-free DMEM for 1 h and replaced with fresh media. Afterward, DiI-loaded PCSNs were added to the medium for another 1 h of incubation. Then, the cells were analyzed with a confocal microscope (Olympus FV1000) connected to a CO$_2$ incubator.

**Animals and tumor models**. Female nude mice (18 ± 2 g, 6 weeks of age) were fed under the condition of 22 ± 2 °C and 55 ± 5% humidity, with free access to food and water. All animal experiments were approved by the Institutional Animal Care and Use Committee at Ulsan National Institute of Science and Technology and conducted in compliance with the guidelines (UNIST-IACUC-17-29). To set up the tumor xenograft model, mice were subcutaneously inoculated in the right lower leg with $1 \times 10^6$ human breast cancer SK-BR3 cells. Tumor volume ($V$) was determined by the following equation: $V = L \times W^2/2$, where $L$ and $W$ are the length and width of the tumor, respectively. SK-BR3 tumor-bearing mice were used in the experiments when the tumor volumes reached approximately 100 mm$^3$.

**In vivo and ex vivo fluorescence imaging**. To evaluate the biodistribution of GSH-MSN, PEG-MSN, and PCSN loaded with a lipophillic dye, DiD (at 10 wt% loading capacity) was intravenously injected (0.1 mg/mL) into SK-BR3 tumor-bearing nude mice (7 mice/group). Then, the mice were anesthetized and imaged using an in vivo imaging system (Bruker Xtreme) with an excitation wavelength of 630 nm and emission wavelength of 700 nm. In vivo images were taken at 0.5, 1, 2, 4, 8, 10, 12, 24, 32, and 48 h. Then, the mice were killed to separate the organs and tumors for ex vivo imaging to determine the biodistribution pattern of DiD-loaded nanocarriers.

**In vivo tumor inhibition analysis**. Female nude mice (average weight 18.5 ± 2 g) were obtained from Orientbio, Korea, for the in vivo studies. All protocols for the in vivo experiments were approved by the UNIST-IACUC animal ethics approval committee. SK-BR3 tumor-bearing mice were used to carry out experiments when the tumor volumes reached approximately 300 mm$^3$. The tumor-bearing nude mice were randomly divided into five groups (6 mice per group). PBS, free-CPT, PCSN, CPT-loaded PEG-MSN, and CPT-loaded PCSN were intravenously injected into mice every 3 days seven times. CPT dosage was 1.5 mg/kg body weight. The first day of treatment was defined as day 0. The body weight and tumor size were recorded every 3 days, and the survival of mice was monitored throughout the

experiment. The concentrations used for the groups treated with PCSN were the same as that of CPT-PCSN. After treatment for 21 days, related mice were killed, and the tumor tissues were removed from the bodies to investigate the morphology and use for further studies.

**Histochemical analysis**. At day 22, one mouse in each group was sacrificed to separate the tumors, and the mice were then fixed in 4% formaldehyde solution in PBS. For histochemical staining, the fixed tumors were dehydrated by gradient ethanol washing and embedded in paraffin blocks, sectioned, and stained with H&E staining.

**Statistical analysis**. All data were reported as means ± standard deviations (SDs). Microsoft Excel software was used to calculate $P$-values for unpaired $t$-tests. Statistical significance was determined by using Student's $t$-test and one-way analysis of variance, and $P$-values of <0.05 were considered to be indicative of statistical significance.

## Data availability

The data that support the findings of this study are available from the corresponding author upon reasonable request.

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

## Acknowledgements

This work was supported by the National Research Foundation of Korea (NRF) Grant funded by the Korean Government (MSIP) (2016R1A5A1009405, 2015H1D3A1061983, 2017R1A2B4003617, 2018R1A6A1A03025810, and 2014R1A5A1009799). We thank the support of UNIST Central Research Facilities (UCRF) for LC-MS/MS data.

## Author Contributions

J.-H.R., S.K., and C.K. conceived, designed, and supervised the study, analyzed and interpreted data, and wrote the manuscript. J.Y.O., H.S.K., and L.P. have equally contributed, designed, carried out and analyzed data from most of the experiments, and wrote the manuscript with input from all co-authors. H.Y.K. and K.K. helped some experiments. B.J. and S.A.P. performed and analyzed some in vivo experiments. E.M.G. and S.K.K. performed molecular simulations. J.K.S. performed and analyzed the proteomic study. All authors discussed the results and commented on the manuscript.

## Additional information

**Competing interests:** The authors declare no competing interests.

