## [Peer Review File · Nature Communications]

Reviewers' comments:

Reviewer #1 (Remarks to the Author):

This work is a very good piece of science. It contains many elements of novelty and will stimulate new studies in the field of bio-nano interactions. Surely the experimental strategy is very clear and well thought out. The body of data is abundant and many experimental techniques are used. Experimental data support the conclusions of the work, but this reviewer believes that some other hypotheses about authors' findings could be made. This could increase the overall significance of the manuscript as well as its impact in the community. In summary, the overall opinion is highly positive and the article should be published after addressing the following minor points.

1) First, the authors investigated the composition of serum proteins adsorbed on each NP using shotgun proteomics, but they did not report any physical-characterization of NP-protein complexes. How are size and zeta-potential distributions affected by interaction with serum properties? Does the PDI change following exposure to serum proteins? Some relevant information could arise from those experiments.

2) The authors ascribe exclusively to the targeting molecule the ability to improve in vitro uptake and to limit tumor growth in vivo. However, some of the corona proteins could play a role in both processes. For example, in panel D of Figure 2 we can see that apolipoproteins are particularly abundant in the corona of PCSNs with respect to GSH-MSN and to PEG-MSN. Some recent studies have shown that apolipoproteins play an important role in association with target cells (see for example: *Nanoscale*, 2017, 9 (44), 17254-17262; *Nanoscale*, 2014, 6, 2782; *Nanoscale*, 2018, 10 (9): 4167-4172). The authors should comment on this point, namely that the optimized performances of PCSNs could be, at least partly, due to the serum proteins recruited at the particle surface. Probably the authors have not emphasized this aspect because the PCSN corona is largely less rich than that of the other particles. However, even a few proteins, if properly exposed, could play a key role in binding with target cells.

3) Similarly, the scarcely abundant corona of PCSNs, by virtue of its specific composition, could give the particles a "stealth" effect. This possibility has been investigated previously in the literature (*Langmuir* 2015, 31 (39), 10764-10773). This point should be discussed and some of the works should be cited .

4) Personally this reviewer is rather skeptical about the CGMD simulation results. Even if HSA is the largely the most abundant protein in serum, an aqueous environment containing only HSA is an oversimplification of the physiological environment seen by NPs . This is demonstrated by the fact that for many nanoparticle formulations the corona composition does not reflect the abundance of proteins in serum. In many cases HSA is not among the most abundant corona proteins. I would suggest removing these results from the body of the article and returning them to additional materials where other simulation results are placed. The readability of the manuscript would undoubtedly benefit from this.

Reviewer #2 (Remarks to the Author):

My comments

- Recently, many authors have characterized and evaluated serum protein adsorption on nanoparticles systems and their behavior in vitro as well as in vivo (some examples: doi: 10.1016/j.biomaterials.2017.01.007, DOI: 10.1016/j.bbagen.2018.03.026, DOI: 10.1039/c7nr08509e, DOI: 10.1021/nn404481f). Present paper describes the HER2-binding affibody combined with GST protein covered on mesoporous silica nanoparticle (MSN). Authors have evaluated their targeting delivery system by appropriate in vitro and in vivo methods.

However, With reference to the recently published literature, current paper does not justify the novelty of the research to be published in this journal. Therefore, I do not recommend this article to be published in the journal.

Few points needs to consider further:

- Fig. 1A is confusing and does not represent the concept of the article clearly.
- What about result of proteins adsorbed after 2 and 4 hours of incubation?
- How can authors control orientation of Afb on the surface of nanoparticles? It can be the key to avoid the serum proteins on the surface of nanoparticles.
- Comparison of in vitro and in vivo data of marketed Camptothecin formulation with the PCSN and PEG-MSN would be more interesting and give more insights on effect of protein corona on uptake of nanoparticles.
- It would have been interesting to see if only HER2-Afb-modified MSNs (without GST, a fusion tag protein) were evaluated in vitro as well as in vivo. Author should mention the role of GST protein in detail either in the introduction or discussion part of the article which would further facilitate the readers.

Reviewer #3 (Remarks to the Author):

General Comments:

In this Communication Oh et al. reported a new scheme of constructing a supramolecular assembly of silica nanoparticle transporter grafted with a HER2-affibody shield, for improved drug delivery and tumor targeting. Overall this study was well designed and comprehensively executed with characterisations in vitro and in vivo. The topic of the research is important and timely for the field of nanomedicine.

However, there are few issues and inconsistencies which are required to be addressed before the acceptance of the manuscript.

Specific comments:

1. Line 21, the authors highlighted the role of thermodynamic stabilization of the orientations of GST-HER2-Afb on silica nanoparticles as a crucial factor for the biocirculation and targeting of the nanoparticles. It was demonstrated by their MD simulations that the stabilization of GST-HER2-Afb on silica nanoparticles was due to specific interaction between GST and GSH. Authors should also include the role of electrostatic interaction between GSH-capped silica nanoparticles and GST-HER2-Afb in the abstract.
2. Line 52, the authors mentioned that PEG and carbohydrate NPs could prevent corona formation. This is inaccurate as PEG and carbohydrate coatings actually also induce plasma protein associations, as discussed extensively in ref 21 and by Wang et al (Small 2017, 13, 1701528). In ref 21, for example, the binding of most enriched protein clusterin was used as an effective strategy against nanoparticle clearance. The authors are therefore advised to mend the partial statement on line 53, that preventing protein corona could prolong the circulation of NPs and escape cell clearance.
3. Line 72, the stability of GST-HER2-Afb was referred to Fig. 1D, however, Fig. 1D was about the stability of PCSN.
4. PCSN looked aggregated in Figure 1F, which should be the reason for their increased hydrodynamic size from 140 to 270 nm. In TEM, PCSN and GSH silica NPs appeared to be of similar sizes. Please comment.
5. It will improve the readability of the paper should the authors enlist the full names of the drugs.

6. Line 71, the authors mentioned that the surface charge of GST-HER2-Afb was similar to that of GST. But nowhere could the reviewer find the charge of GST in the paper.
7. Fig. 1, panel G was not labelled but was referred to in the text.
8. Line 111, the author stated that GSH-MSN was well stabilized, but the size change was ~50%. Please comment.
9. Line 164, "Fig. 2E left" should be "Fig. 2E right", and "Fig. 2E right" should be "Fig. 2E left".
10. Line 181, the abbreviation of PDS, likely denoting polydioxanone, was not introduced.
11. Why the use of HER2-binding affibody, not affibodies for other types of proteins? This needs to be justified in the Introduction section.
12. The use of supramolecular assembly for prolonged nanoparticle biocirculation and drug delivery is not a new concept as the authors claimed in the abstract and introduction, even though this particular design is new. The concept was first demonstrated by Kah et al. (*ACS Nano*, 2012, 6, 6730) and outlined in the Perspective by Ke et al. (*ACS Nano*, 2017, 11, 11773). These papers should be acknowledged.
13. There were numerous grammar issues in the SI, especially in the Methods section. The main text was written clearly in contrast.

Reviewer #1 (Remarks to the Author):

This work is a very good piece of science. It contains many elements of novelty and will stimulate new studies in the field of bio-nano interactions. Surely the experimental strategy is very clear and well thought out. The body of data is abundant and many experimental techniques are used. Experimental data support the conclusions of the work, but this reviewer believes that some other hypotheses about authors' findings could be made. This could increase the overall significance of the manuscript as well as its impact in the community. In summary, the overall opinion is highly positive and the article should be published after addressing the following minor points.

1) First, the authors investigated the composition of serum proteins adsorbed on each NP using shotgun proteomics, but they did not report any physical-characterization of NP-protein complexes. How are size and zeta-potential distributions affected by interaction with serum properties? Does the PDI change following exposure to serum proteins? Some relevant information could arise from those experiments.

Answer: We appreciate the reviewer's thoughtful comments. As the reviewer mentioned, we further studied physical-characterization of the NP-protein complexes and the relevant information was added in the manuscript.

The GSH-MSN, PEG-MSN, and PCSN were treated with 55% serum for 1h, 2h, and 4h, respectively, and washed with PBS three times by centrifugation (at 5000 rpm for 3 minutes). Each nanoparticle was re-dispersed in PBS and measured its size, surface charge, and PDI by Zetasizer Nano ZS. The size of GSH-MSN (140 ± 20 nm) and PEG-MSN (190 ± 20 nm) increased by a factor of 2.5~3, when treated with 55% serum for 1h, 2h, and 4 h, respectively, indicating a protein corona formation. On the other hand, the size of PCSN ($\sim 270 \pm 20$ nm) treated with 55% serum for 1h, 2h, and 4 h was 285 ± 20 nm, 252 ± 20 nm, and 290 ± 20 nm, respectively, showing no significant change. Similarly, the surface charge of nanoparticle was changed from -40 ± 3 mv to -7 ± 1 mv for GSH-MSN and from -1 ± 0.1 mv to -5 ± 1 mv for PEG-MSN when treated with 55% serum, which is presumably due to the screening of the serum proteins. Meanwhile, the change of the surface charge for PCSN was rather negligible. In the case of the PDI, the exposure of the serum did not show significant change for all nanoparticles. Overall, in line with the SDS-PAGE results, the physical-characterization results confirmed the decreased PCSN-serum protein interactions. As the reviewer's suggestion, the discussion (**line 8-12 on page 6**) and additional figure (**Supplementary Fig. 5**) about this aspect with relevant references was added in the manuscript as follows.

“To further confirm the particle-protein interactions, the physical-characterizations were investigated by using DSL, zeta, and PDI (**Supplementary Fig. 5**). The treatment of 55% serum on GSH-MSN and PEG-MSN increased its hydrodynamic radius and changed its surface charge while PCSN showed no significant changes, which confirms the decreased PCSN-serum protein interactions.”

Supplementary Fig. 5. The physical-characterizations of nanoparticle-serum protein interaction. Size, surface charge and PDI analysis of A) GSH-MSN B) PEG-MSN C) PCSN with treatment of the 55% serum for 1h, 2h and 4h. The GSH-MSN, PEG-MSN, and PCSN were treated with 55% serum for 1h, 2h, and 4h, respectively, and washed with PBS three times by centrifugation (at 5000 rpm for 3 minutes). Each nanoparticle was re-dispersed in PBS and measured its size, surface charge, and PDI by Zetasizer Nano ZS. The size of GSH-MSN (140 ± 20 nm) and PEG-MSN (190 ± 20 nm) increased by a factor of 2.5~3, when treated with 55% serum for 1h, 2h, and 4 h, respectively, indicating a protein corona formation. On the other hand, the size of PCSN ($\sim 270 \pm 20$ nm) treated with 55% serum for 1h, 2h, and 4 h was 285 ± 20 nm, 252 ± 20 nm, and 290 ± 20 nm, respectively, showing no significant change. Similarly, the surface charge of nanoparticle was changed from -40 ± 3 mv to -7 ± 1 mv for GSH-MSN and from -1 ± 0.1 mv to -5 ± 1 mv for PEG-MSN when treated with 55% serum, which is presumably due to the screening of the serum proteins. Meanwhile, the change of the surface charge for PCSN was rather negligible. In the case of the PDI, the exposure of the serum did not show significant change for all nanoparticles.

2) *The authors ascribe exclusively to the targeting molecule the ability to improve in vitro uptake and to limit tumor growth in vivo. However, some of the corona proteins could play a role in both processes. For example, in panel D of Figure 2 we can see that apolipoproteins are particularly abundant in the corona of PCSNs with respect to GSH-MSN and to PEG-MSN. Some recent studies have shown that apolipoproteins play an important role in association with target cells (see for example: Nanoscale, 2017, 9 (44), 17254-17262; Nanoscale, 2014, 6, 2782; Nanoscale, 2018, 10 (9): 4167-4172). The authors should comment on this point, namely that the optimized performances of PCSNs could be, at least partly, due to the serum proteins recruited at the particle surface. Probably the authors have not emphasized this aspect because the PCSN corona is largely less rich than that of the other particles. However, even a few proteins, if properly exposed, could play a key role in binding with target cells.*

Answer: We appreciate the reviewer's thoughtful comment. Based on our observation from the shotgun proteomics and SDS-PAGE, we could conclude that the improved targeting and therapeutic efficacy of PCSN in vitro and in vivo was due to the reduced serum protein absorption (in specific immune response involving proteins (e.g. complement, and coagulation proteins)) on PCSN. In addition, as the reviewer pointed out, some studies reported that specific apolipoproteins play an important role in association with target cells, although we somewhat overlooked this point. We agree with the reviewer's comment, so the discussion about this aspect with relevant references was added in the manuscript (line 27-31 on page 6) as follows.

“It is also noted that the relative quantity of apolipoproteins among serum proteins absorbed on PCSN increased with respect to GSH-MSN and to PEG-MSN (Fig. 2D middle). Considering that apolipoproteins on nanoparticle play an important role in association with cellular uptake to target cells,(33-35) targeting capability of PCSN partly could be linked with some serum proteins recruited at the surface of the PCSN.”

3) *Similarly, the scarcely abundant corona of PCSNs, by virtue of its specific composition, could give the particles a "stealth" effect. This possibility has been investigated previously in the literature (Langmuir 2015, 31 (39), 10764-10773). This point should be discussed and some of the works should be cited.*

Answer: We also thanks to the reviewer for the comment. According to the literature, it was well investigated that the biomolecular corona formation with a specific composition could provide a stealth effect on the macrophage recognition. When incubated with human plasma, the formation of NP-biomolecular coronas retaining immunoglobulins, complement factors, and coagulation proteins could enhance phagocytosis by immune cells. On the other hand, the cellular uptake of the protein pre-coated NP was significantly reduced by macrophage due to the impairment of the relevant functional motifs via screening by other protein corona components. This study well suggests that a specific composition of the biomolecular corona could confer the stealth effect on the particles. Accordingly, as the reviewer's suggestion, the reference and discussion about the effects of the corona composition on the stealth effect was added in the manuscript (line 4-8 on page 9) as follows.

“A recent study reported that the biomolecular corona formation with a specific composition provides a stealth effect on the macrophage recognition (41). Similarly, our results suggest that the scarce corona formation on PCSN caused by the supramolecularly pre-coated proteins could confer stealth properties to evade immune cells and subsequent elimination by the MPS.”

4) *Personally this reviewer is rather skeptical about the CGMD simulation results. Even if HSA is the largely the most abundant protein in serum, an aqueous environment containing only HSA is an oversimplification of the physiological environment seen by NPs. This is demonstrated by the fact that for many nanoparticle formulations the corona composition does not reflect the abundance of proteins in serum. In many cases HSA is not among the most abundant corona proteins. I would suggest removing these results from the body of the article and returning them to additional materials where other simulation results are placed. The readability of the manuscript would undoubtedly benefit from this.*

Answer: We thank the reviewer for the valuable comment and suggestion. As the reviewer commented, the most abundant protein, attached to nanoparticles, was varied depending on the size and coated functional groups of nanoparticles. (Stefan Tenzer, Dominic Docter, Jorg Kuharev, Anna Musyanovych, Verena Fetz, Rouven Hecht, Florian Schlenk, Dagmar Fischer, Klytaimnitra Kiouptsi, Christoph Reinhardt, Katharina Landfester, Hansjorg Schild, Michael Maskos, Shirley K. Knauer and Roland H. Stauber, Rapid formation of plasma protein corona critically affects nanoparticle pathophysiology, *Nat. Nanotechnol.*(8, 772-781, 2013. DOI: 10.1038/ NNANO.2013.181) The reason for using albumin in this CGMD simulation, which focused to investigate the interaction between serum protein and PCSN series (PEG-MSN, PCSN(R), PCSN(-), PCSN), was that albumin occupied the highest percentage of attached protein ‘on PCSN series’ in the experimental result (Fig. 2E).

In this paper, the total number of attached protein on nanoparticle was 183 and 78 of the most abundantly attached protein was shown in Fig. 2. In Fig. 2E, which presented the amount of attached protein on PCSN series by SDS-PAGE, dark line was appeared at the 78 kDa part, which is the mass of albumin. Therefore, we confirmed that the protein attached to PCSN series was mainly albumin. Additionally, what we intended to explain with CGMD simulation results was that comparison the interaction with external proteins, according to coated materials and structure of coating on corona NP. Consequently, albumin, which was shown lot of attachment in common, was used for external protein.

In short, the CGMD results could explain the reason that the PCSN was more advantageous than other NP models for preventing external protein adsorption by comparing the interaction of serum protein (albumin) with PEG-MSN, PCSN(R), PCSN(-) and PCSN. However, as the reviewer scientifically pointed out, albumin did not reflect an environment including all types of proteins, thus, we moved the simulation results into Supplementary Information for clear readability.

Also, the reason for using albumin was not clear, we modified the sentence in the Supplementary Information (page 10) as follows.

“Albumin was used to investigate the interactions between coated silica nanoparticles and serum protein because it was the most abundantly adsorbed group of serum proteins with coated silica nanoparticles (Fig. 2E).”

Reviewer #2 (Remarks to the Author):

My comments

Recently, many authors have characterized and evaluated serum protein adsorption on nanoparticles systems and their behavior in vitro as well as in vivo (some examples: doi: 10.1016/j.biomaterials.2017.01.007, DOI: 10.1016/j.bbagen.2018.03.026, DOI: 10.1039/c7nr08509e, DOI: 10.1021/nn404481f). Present paper describes the HER2-binding affibody combined with GST protein covered on mesoporous silica nanoparticle (MSN). Authors have evaluated their targeting delivery system by appropriate in vitro and in vivo methods. However, with reference to the recently published literature, current paper does not justify the novelty of the research to be published in this journal. Therefore, I do not recommend this article to be published in the journal.

Answer: We thank to the reviewer for the comment. As the reviewer pointed out, some studies about characterizations of protein corona on nanoparticles systems and their behaviors in vitro and in vivo have already been reported. For examples, D'Hollander et al (doi: 10.1016/j.biomaterials.2017.01.007) demonstrated that protein corona formation on gold nanoparticle could be controlled by modifying chemical functional groups. Kah et al (doi: 10.1039/c7nr08509e) reported a serum protein-stabilized gold nanorod that could be used for drug delivery carrier. Grandori et al (doi: 10.1016/j.bbagen.2018.03.026) reported that conformational changes of native protein structure certainly occur when absorbed on nanoparticle. Furthermore, Caruso et al (doi: 10.1021/nn404481f) demonstrated that a conformational change of protein corona on particle could affect the cellular uptake of the nanoparticles. *Besides these examples, some literatures about protein corona have been published, but a deeper understanding of the molecular mechanism about the nanoparticle–protein corona interactions still remains largely unknown and furthermore supramolecularly regulating corona formation on targeted drug delivery system have not been reported yet. There is no report of a nanoparticle system which minimizes interactions with serum proteins to prevent the clearance of these particles by macrophages, while ensuring their targeting function in vitro and in vivo .*

In this respect, we here developed a nanoparticulate system with a protein corona shield in which the recombinant fusion protein (GST-affibody) was formed on the particle via enzyme-substrate specificity. *We demonstrated that the supramolecularly forming corona could be a key for stabilizing the particle, reducing additional interaction with serum proteins, and eventually enhancing targeting and therapeutic efficacy of the particle in vitro and in vivo.* We used many experimental techniques and surely supplied enough data to draw the conclusion. This finding can provide a new targeting platform for the biomedical community since numerous functional proteins can be installed by the similar fashion. Thus, we believe that our works contain many elements of novelty and make an impact on the bio-nano scientific community.

Few points needs to consider further:

- *Fig. 1A is confusing and does not represent the concept of the article clearly.*

Answer: As the reviewer pointed out, the scheme in Fig. 1A is modified to clarify the concept of the study and replaced in the manuscript.

• *What about result of proteins adsorbed after 2 and 4 hours of incubation?*

Answer: The results of proteins adsorbed on each particle after 2 and 4 hours of serum incubation exhibited negligible changes comparing with the results of 1 h serum treatment, which was confirmed by SDS-PAGE (Fig. 2B) and physical-characterizations obtained from additional experiments (Supplementary Fig. 5). In the case of the protein analysis, we run the experiments for each particle treated 1h serum incubation because it is generally recognized that a stable composition is achieved in one hour according to literatures (doi: 10.1021/la401192x and 10.1021/ja910675v).

• *How can authors control orientation of Afb on the surface of nanoparticles? It can be the key to avoid the serum proteins on the surface of nanoparticles.*

Answer: As the reviewer mentioned, controlling orientation of GST-HER2-Afb on nanoparticle can be the key to avoid the serum proteins. To show this, we utilized both noncovalent (oriented) and covalent (randomly oriented) conjugation for the GST-HER2-Afb on GSH-MSN. In specific, for the non-covalent (oriented) conjugation, the surface of the nanoparticle was modified with GSH and GST-HER2-Afb containing GSH binding site was non-covalently attached on the particle via enzyme-substrate specific interaction. In contrast, for the covalent (randomly oriented) conjugation, GST-HER2-Afb was covalently attached on succinimidyl-modified MSN by amide formation with amine groups on the protein surface. We used SDS-PAGE (Fig. 2E left) to assess the effect of the orientation of GST-HER2-Afb and protein absorption on PCSN(R) (randomly oriented) was enhanced 5-fold compared with that for PCSN (oriented). The results indicate that the randomly orientated conjugation of GST-HER2-Afb may change surface properties and increase protein–nanoparticle interactions.

• *Comparison of in vitro and in vivo data of marketed Camptothecin formulation with the PCSN and PEG-MSN would be more interesting and give more insights on effect of protein corona on uptake of nanoparticles.*

Answer: We thank to the reviewer for the comment. As the reviewer suggested, we examined therapeutic efficacy of the marketed formulation for Camptothecin. We prepared a marketed Camptothecin formulation (CPT-loaded Cremophor EL) by using Cremophor EL that is clinically used formulation vehicle for water-insoluble anticancer agents. The CPT-loaded Cremophor EL (1.5mg/kg (CPT basis) were intravenously injected to xenograft tumor established by using SKBR3 cell in nude mice. The xenograft tumor was established by using human breast cancer cell line (SKBR3) in nude mice. The mice were separated in two group (control and drug) having six mice in each group, when the tumor size reaches $\sim 200 \text{ mm}^3$. Then, CPT-loaded Cremophor EL (1.5mg/kg (CPT basis) and 1X PBS were injected intravenously every alternative day up-to three weeks. The tumor volume, tumor weight, and the body weight were measured in all the group in indicated days and compared. The result indicates that there is no significant change of tumor size and tumor weight after the drug treatment (Fig. R1), suggesting a negligible antitumor effect of the CPT-loaded Cremophor EL.

Fig. R1. *In vivo* antitumor study of drug in SKBR3 xenograft model. (a) Tumor volume of drug treated and control mice in indicative day. (b) No significant change in the body weight of mice was observed after drug treatment. (c) Tumor image and (d) tumor weight of drug treated and control mice. The result indicates that the drug does not have the antitumor effect.

• It would have been interesting to see if only HER2-Afb-modified MSNs (without GST, a fusion tag protein) were evaluated *in vitro* as well as *in vivo*. Author should mention the role of GST protein in detail either in the introduction or discussion part of the article which would further facilitate the readers.

Answer: We thank to the reviewer for the comment. As the reviewer suggest, we evaluated the targeting efficacy of the HER2-Afb-modified MSNs (without GST, a fusion tag protein). We generated a new HER2-Afb and MSNs was further modified with HER2-Afb through EDC-NHS reaction to give the HER2-Afb-modified MSNs. To investigate the targeting efficacy, dye-containing (DiIC₁₈) HER2-Afb-modified MSN was incubated in solutions containing 10% and 55% of serum, respectively, and then treated with SK-BR3 cells. From the confocal microscopy images, we confirmed that cellular uptake for HER2-Afb-modified MSN was reduced and rather stuck outside of the cells, which is presumably due to the screening effect of corona formation on the randomly oriented HER2-Afb of particle (Supplementary Fig. 14). These results suggest that GST plays an important role in reducing interactions among the serum proteins as well as supramolecularly conjugating on particle. As the reviewer's suggestion, the discussion and an additional figure (Supplementary Fig. 14) was added in the manuscript (line 10-15 on page 10) as follows.

“A control, HER2-binding affibody (Afb) modified MSNs (Afb-MSN) was also evaluated and it confirmed that GST plays an important role in reducing interactions among the serum proteins as well as supramolecularly conjugating on particle (Supplementary Fig. 14).”

Supplementary Fig. 14. Cellular uptake experiment for dye-containing (DiIC₁₈) HER2-binding affibody (Afb) modified MSNs (Afb-MSN) on SK-BR3 cell. The Afb-MSN was pre-treated in a) 10% serum and b) 55% serum for 1 hour. We generated a new HER2-Afb and MSNs was further modified with HER2-Afb through EDC-NHS reaction to give the HER2-Afb-modified MSNs. To investigate the targeting efficacy, dye-containing (DiIC₁₈) HER2-Afb-modified MSN was incubated in solutions containing 10% and 55% of serum, respectively, and then treated with SK-BR3 cells. From the confocal microscopy images, we confirmed that cellular uptake for HER2-Afb-modified MSN was reduced and rather stuck outside of the cells, which is presumably due to the screening effect of corona formation on the randomly oriented HER2-Afb of particle (Fig. S14). These results suggest that GST plays an important role in reducing interactions among the serum proteins as well as supramolecularly conjugating on particle.

Reviewer #3 (Remarks to the Author):

General Comments:

In this Communication Oh et al. reported a new scheme of constructing a supramolecular assembly of silica nanoparticle transporter grafted with a HER2-affibody shield, for improved drug delivery and tumor targeting. Overall this study was well designed and comprehensively executed with characterizations in vitro and in vivo. The topic of the research is important and timely for the field of nanomedicine.

However, there are few issues and inconsistencies which are required to be addressed before the acceptance of the manuscript.

Specific comments:

1. Line 21, the authors highlighted the role of thermodynamic stabilization of the orientations of GST-HER2-Afb on silica nanoparticles as a crucial factor for the biocirculation and targeting of the nanoparticles. It was demonstrated by their MD simulations that the stabilization of GST-HER2-Afb on silica nanoparticles was due to specific interaction between GST and GSH. Authors should also include the role of electrostatic interaction between GSH-capped silica nanoparticles and GST-HER2-Afb in the abstract.

Answer: We appreciate the reviewer's thoughtful comment. We emphasized that the thermodynamically stabilization of GST-HER2-Afb in an orientated way on silica nanoparticles could be a crucial factor for the biocirculation and targeting of the nanoparticles. Besides, as the reviewer mentioned, the electrostatic interaction between GST-HER2-Afb and serum proteins can be a factor for enhanced therapeutic efficacy for the nanoparticle. We agree with the reviewer's point and it was added in the manuscript (line 21-23 on page 8) as follows.

“which suggests that the orientation of GST-HER2-Afb attached to the surface of silica nanoparticles as well as the electrostatic interactions can be an important factor.”

2. Line 52, the authors mentioned that PEG and carbohydrate NPs could prevent corona formation. This is inaccurate as PEG and carbohydrate coatings actually also induce plasma protein associations, as discussed extensively in ref 21 and by Wang et al (Small 2017, 13, 1701528). In ref 21, for example, the binding of most enriched protein clusterin was used as an effective strategy against nanoparticle clearance. The authors are therefore advised to mend the partial statement on line 53, that preventing protein corona could prolong the circulation of NPs and escape cell clearance.

Answer: We thank you for the comment. As the reviewer pointed out, nanoparticles with PEG and carbohydrate coatings can induce some specific plasma protein associations such as the clusterin, so the nanoparticles can enhance the circulation time and MPS clearance. Therefore, the description about protein corona formation that “can be prevented” was modified to “can be regulated” in the manuscript (line 19-23 on page 2) as follows.

“The formation of a protein corona on the nanoparticle surface can be regulated by modifying the nanoparticle surface with zwitterions, PEG, carbohydrate moieties, and dysopsonic proteins, which can enhance the colloidal stability and prolong the circulation time in blood by enabling escape from MPS clearance”

3. Line 72, the stability of GST-HER2-Afb was referred to Fig. 1D, however, Fig. 1D was about the stability of PCSN.

Answer: As the reviewer pointed out, the Fig. 1D is showing the stability of the PCSN, so the description of the stability of GST-HER2-Afb in the line 72 was removed from the manuscript.

4. PCSN looked aggregated in Figure 1F, which should be the reason for their increased hydrodynamic size from 140 to 270 nm. In TEM, PCSN and GSH silica NPs appeared to be of similar sizes. Please comment.

Answer: As the reviewer mentioned, the size of the nanoparticles appeared to be similar in TEM image but different in the DLS results. The nanoparticle samples for DLS were in a solvated state where there could be solvent molecules dynamically associated with the particles while the nanoparticle samples for TEM were in a dry state, exhibiting the most compact state. So, for the DLS results, the difference of the hydrodynamic size between PCSN and GSH silica NPs is presumably due to dynamic interactions of the non-covalently pre-coated GST-HER2-Afb of PCSN with each other or water molecules in a solvated state. Additionally, to avoid confusion for analyzing the TEM image of PCSN, it was replaced by another one.

5. It will improve the readability of the paper should the authors enlist the full names of the drugs.

Answer: As the reviewer suggested, the full name of the drugs was enlisted in the manuscript.

6. Line 71, the authors mentioned that the surface charge of GST-HER2-Afb was similar to that of GST. But nowhere could the reviewer find the charge of GST in the paper.

Answer: We thank you for the comment. As the reviewer pointed out, the surface charge of the GST was missed. The surface charge measurement of the GST was added in the Fig. 1C.

7. Fig. 1, panel G was not labelled but was referred to in the text.

Answer: The “Fig. 1G” was replaced to “Fig. 1F (left)”.

8. Line 111, the authored stated that GSH-MSN was well stabilized, but the size change was ~50%. Please comment.

Answer: As the reviewer pointed out, we stated that “No significant aggregation of PCSNs in PBS was observed in the DLS analysis for up to 2 weeks at 4 °C, indicating that the GSH-MSN was well stabilized by the GST-HER2-Afb” The description was to highlight the colloidal stability of PCSN due to the pre-coating of GST-HER2-Afb in aqueous solution up to 2 weeks because the diameter difference of the PCSN before and after 2 weeks exhibited around 35nm. To clarify it, the statement was modified (line 28-29 on page 5) as follows.

“No significant aggregation of PCSNs was observed in the DLS analysis up to 2 weeks at 4 °C (Fig. 1D).”

9. Line 164, “Fig. 2E left” should be “Fig. 2E right”, and “Fig. 2E right” should be “Fig. 2E left”.

Answer: Those were corrected as the reviewer pointed out.

10. Line 181, the abbreviation of PDS, likely denoting polydioxanone, was not introduced.

Answer: The full form of PDS in the molecular simulation section is now introduced as “pyridine disulfide hydrochloride” in the Supporting Information that was removed from main text according to the other reviewer.

11. Why the use of HER2-binding affibody, not affibodies for other types of proteins? This needs to be justified in the Introduction section.

Answer: Besides HER2-binding affibody, we also examined the versatile targeting platform of the protein corona shield by using EGFR-binding affibody. The EGFR-binding affibody was conjugate to glutathione-S-transferase to give GST-EGFR-Afb. When applied on the GSH-MSN, similar to PCSN (GST-HER2-Afb modified MSN), its targeting ability to MDA-MB468 cells (EGFR-positive cancer cells) was observed. The relevant data is placed in **Supplementary Fig. 15** and the justification of the versatile targeting platform using affibodies is described in the manuscript (**line 13-16 on page 10**) as follows.

“To examine the versatile targeting platform for PCS, EGFR-binding Afb combined with glutathione-S-transferase (GST-EGFR-Afb) was also applied on GSH-MSN and its targeting ability to MDA-MB468 cells (EGFR-positive cancer cells) was confirmed (**Supplementary Fig. 15**). Taken together, these results show that PCSN exhibited cell-specific targeting ability as well as stealth properties.”

12. The use of supramolecular assembly for prolonged nanoparticle biocirculation and drug delivery is not a new concept as the authors claimed in the abstract and introduction, even though this particular design is new. The concept was first demonstrated by Kah et al. (*ACS Nano*, 2012, 6, 6730) and outlined in the Perspective by Ke et al. (*ACS Nano*, 2017, 11, 11773). These papers should be acknowledged.

Answer: We thank the reviewer for the comment. As the reviewer mentioned, there already are some reports about drug delivery nanoparticles that were encapsulated by protein corona via non-covalent interaction (supramolecular assembly). We agree with the reviewer’s point, so the papers were added in the Reference section (**reference 23 and 24**).

13. There were numerous grammar issues in the SI, especially in the Methods section. The main text was written clearly in contrast.

Answer: The SI was carefully reviewed by an experienced editor whose first language is English and the grammar issue and typo were corrected.

REVIEWERS' COMMENTS:

Reviewer #1 (Remarks to the Author):

All the points I raised have been properly addressed by the Authors. The paper should be published in its present form

Reviewer #2 (Remarks to the Author):

Reviewer's comment: Authors have suitably justified the comments raised by the reviewer by further experiments and new data which clarifies the novelty issue of the article. Figure 1 has been modified properly and improves the understanding of the figure for the readers.

Grammar and spell check need to be done thoroughly before publishing this article.

Reviewer #3 (Remarks to the Author):

The authors have incorporated this reviewer's comments adequately. The manuscript is recommended for acceptance by Nat Comm.

REVIEWERS' COMMENTS:

Reviewer #1 (Remarks to the Author):

Reviewer's comment: All the points I raised have been properly addressed by the Authors. The paper should be published in its present form

We really appreciate your review of our paper. Also we thank you for giving us the opportunity to improve our paper in a better way.

Reviewer #2 (Remarks to the Author):

Reviewer's comment: Authors have suitably justified the comments raised by the reviewer by further experiments and new data which clarifies the novelty issue of the article. Figure 1 has been modified properly and improves the understanding of the figure for the readers.

We really appreciate your review of our paper. we thank you for giving us the opportunity to improve our paper in a better way. We also checked the overall spelling and grammar as suggested by the reviewer.

We sincerely thank the reviewer for his/her comments. Yes, we will perform them accordingly.

Reviewer #3 (Remarks to the Author):

The authors have incorporated this reviewer's comments adequately. The manuscript is recommended for acceptance by Nat Comm.

We really appreciate your review of our paper. Also we thank you for giving us the opportunity to improve our paper in a better way.